# Statistical privacy-preserving message broadcast for peer-to-peer networks

**David Mödinger**[1]*, **Jan-Hendrik Lorenz**[2], **Franz J. Hauck**[1]

**1** Institute of Distributed Systems, Ulm University, Ulm, Germany, **2** Institute of Theoretical Computer Science, Ulm University, Ulm, Germany

* david.moedinger@uni-ulm.de

## Abstract

Privacy concerns are widely discussed in research and society in general. For the public infrastructure of financial blockchains, this discussion encompasses the privacy of the originator of a transaction broadcasted on the underlying peer-to-peer network. Adaptive diffusion is an approach to expose an alternative source of a message to attackers. However, this approach assumes an unsuitable attacker model and a non-realistic network model for current peer-to-peer networks on the Internet. We transform adaptive diffusion into a new statistical privacy-preserving broadcast protocol for realistic current networks. We model a class of unstructured peer-to-peer networks as organically growing graphs and provide models for other classes of such networks. We show that the distribution of shortest paths can be modelled using a normal distribution $\mathcal{N}(\mu, \sigma^2)$. We determine statistical estimators for $\mu$, $\sigma$ via multivariate models. The model behaves logarithmic over the number of nodes $n$ and proportional to an inverse exponential over the number of added edges per node $k$. These results facilitate the computation of optimal forwarding probabilities during the dissemination phase for maximum privacy, with participants having only limited information about network topology.

**Data Availability Statement:** All data and materials are available here: https://github.com/vs-uulm/eta-adaptive affiliated software is available under an open source license.

## 1 Introduction

An increasing number of data breaches and media coverage of privacy concerns has led to a heightened awareness of privacy concerns in research and for laypersons. Especially in financial contexts, such as cryptocurrencies, engineers and researchers produced many privacy-improving proposals—either improving privacy on otherwise non-privacy–preserving systems [1] or implementing new systems with privacy-first practices [2–4]. Financial transactions are anonymised in these systems, but they still require broadcasting to all participants.

Privacy for network message broadcasts is also relevant outside of cryptocurrencies. Peer-to-peer file-sharing or content networks [5] such as Gnutella [6] or IPFS [7] use broadcasts, or behaviour similar to a broadcast, for search queries. These queries can be used to track user behaviour even for very sensitive data, such as health conditions and personal interests.

Broadcast behaviour provides additional challenges for privacy [8–12], as the message must be available to all participants. Attempts to improve privacy of peer-to-peer networks include

**Funding:** The authors received no specific funding for this work.

**Competing interests:** The authors have declared that no competing interests exist.

established protocols such as Tor or I2P [13], as well as new protocols [14–17]. These new protocols are tailor-made for broadcast applications, which was not a goal in classical protocols such as Tor.

Our previous proposal [16, 18] considered adaptive diffusion [19] as an intermediate privacy providing phase. Unfortunately, the attacker model of adaptive diffusion is based on a snapshot knowledge of nodes that already received a given message. This model is not well suited for privacy in real-world computer networks, as it does not represent the capabilities of common attackers well, e.g., link information and compromised or cooperating network participants. Further, the forwarding probabilities are derived from an infinite tree network, which is not encountered in real-world networks, as even structured peer-to-peer networks [5] usually contain cycles.

### 1.1 Contributions

In this paper, we clear these obstacles to adoption in real-world computer networks. We transform adaptive diffusion into a protocol for realistic networks. In detail, we

(i). derive optimal forwarding probabilities for adaptive diffusion, based on the abstract distribution of shortest paths in the underlying network,

(ii). model the distribution of shortest paths in $k$-growing graphs,

(iii). provide an estimator for the distribution of shortest paths based on the number of participants $n$ and edges per node $k$, and

(iv). change the protocol so that it can withstand a more realistic attacker model.

### 1.2 Roadmap

In Section 2, we describe the scenario of this paper, as well as relevant background information, including the original adaptive diffusion protocol. Section 3 gives an overview of the resulting transformation of adaptive diffusion. In Sections 4 and 5 we discuss the details of the required changes to the protocol. Section 4 considers the changes in privacy and network assumptions of adaptive diffusion for the transformation to a network protocol. In Section 5, we investigate distributions to determine a concrete implementation of the probabilities involved with the protocol. To achieve this, we derive an estimation of the shortest paths for networks following a k-growing model. Section 6 discusses the privacy properties of the resulting protocol.

## 2 Background

### 2.1 Notation

A short overview of common notation used within this paper is given by Table 1.

**Table 1. Notation used throughout this paper.**

| Notation | Explanation |
| --- | --- |
| $m$ | A given message within our system. |
| $\mathcal{H}(m)$ | A hash function $\mathcal{H}$ applied on a message. |
| $v$ | A vertex of the graph, also called a node or participant of the network. |
| $N(v)$ | Set of all neighbours of node $v$. |
| $\mathfrak{N}_m$ | Selected neighbours of a node during a protocol run for message $m$. |
| $\mathcal{N}(\mu, \sigma^2)$ | Normal distribution with parameters $\mu$ and $\sigma^2$. |

## 2.2 Scenario

In this paper, we discuss the privacy of broadcasts within an unstructured peer-to-peer network. For some applications, e.g., broadcasts of financial transactions in a blockchain network, the sender of a broadcast message has an interest in not being revealed. This is, despite the main goal being everyone receiving their message. The goal is, therefore, to hide the originator of such a message.

The default solution to broadcasting a message in an unstructured network is a flood-and-prune broadcast. Here, the sender sends the message to all its neighbours. A node that has not received the message yet will send it to all of its neighbours. The node excludes the link over which it received the message. Broadcasting, in this way, produces a highly symmetrical dissemination pattern, leading to possible identification attacks.

Assume there are nodes, which are collaborating to identify the originator of such a message, as shown in Fig 1. Those nodes might be distributed throughout the network and can learn the topology of the network over time. These nodes can reliably estimate the identity of the sender of the message by determining the graph centre, or Jordan centre, of the nodes that already received the message. The Jordan centre of the graph is the node, which has the smallest distance to all affected nodes. Given a graph $G$ with a set of vertices $V$ and edges $E$, as well as a distance function $d$, the Jordan centre can be defined as:

$$\text{center}(G_{V,E}) := \text{argmin}_{u \in V} \max(d(u,v) : v \in V). \tag{1}$$

In our scenario, the affected nodes are those that already received the message.

While varying network latencies might distort the result, the set of likely originators is small. Lastly, an attacker might also create connections to all nodes, always receiving the message as a neighbour of the true source.

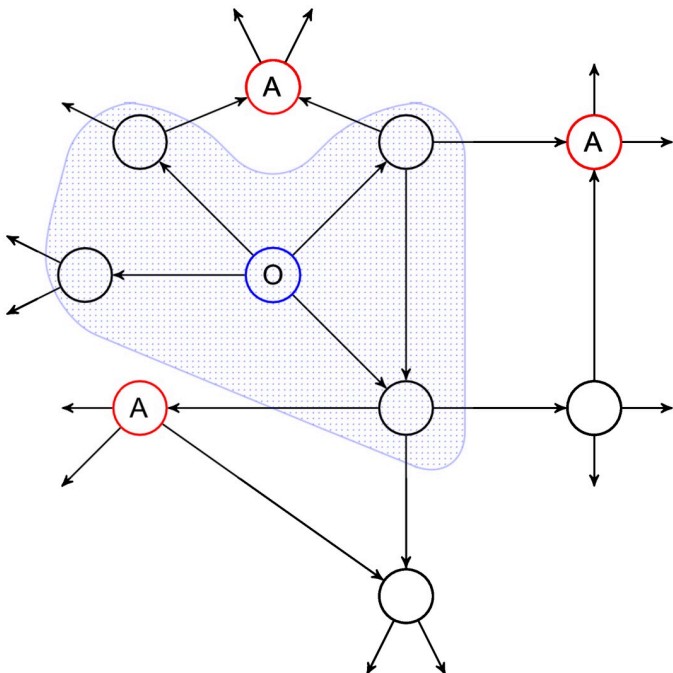

**Fig 1. Motivating example: The originator of a message (O, blue) disseminates a message along the connections (arrows).** The area highlights all nodes that received the message so far. Attackers (A, red) distributed throughout the network will receive the message in the next step and can reconstruct the originating node.

## 2.3 Networks and graphs

Peer-to-peer networks can be constructed in various ways [5]. One of the broadest distinctions of peer-to-peer networks is between structured and unstructured networks. Structured peer-to-peer networks tightly control their overlay, while unstructured peer-to-peer networks have peers join the network on loose rules. Unstructured networks often use broadcasts, often called flooding in peer-to-peer contexts, across the overlay [5].

One set of rules to create such a network has a new peer connect to nodes selected randomly from a list of known participants. This list can be initially retrieved by publicly known participants or via a gossip protocol while part of the network. The number of created connections maintained by new nodes is often fixed in the client source code. Examples of this construction are the network of Bitcoin [20] and classic Gnutella [6].

To model this behaviour via graphs, we use an establishing algorithm. Let us try to establish a network of $n$ nodes, or vertices, by successively adding nodes. Each node establishes $k$ edges, or connections, to previously existing nodes. No node establishes loops, i.e., edges with itself or multiple edges. The result is, therefore, a simple graph. In this paper, we call this a k-growing graph with $n$ nodes.

The previous design ensures a connected component of all network participants. Further, the design is resistant to churn, the act of peers joining and leaving the network, which is not reflected in the model. Churn within a model is a complicated parameters [21], as churn rates may differ for nodes dependent on their network participation, i.e., long-running nodes are often less likely to leave the network.

To introduce adaptive diffusion, we also require the concept of an infinite $d$-regular tree. Such a graph has no cycles and is connected, i.e., between any two nodes exists exactly one path. Further, each node has a degree of $d$, i.e., each node is connected to exactly $d$ neighbours via $d$ edges. As it is infinite, there is no number $n$ limiting the number of nodes and no maximum distance within the graph, often called a graph diameter. For a more in-depth introduction to graphs, c.f. Jackson's Social and Economic Networks [22].

## 2.4 Attacker model

Throughout this paper, we consider attacks on the privacy of network participants. This attack is performed by a colluding fraction of participants of the network, which act in a semi-honest or honest-but-curious manner. Attackers follow the protocol but will attempt to infer the identity of the originator of a given message, i.e., which node created it. This model is used in similar settings [17], as it focuses on information leaks within the protocol.

## 2.5 Probability distributions

For this paper, we revisit some statistical fundamentals in the form of probability distributions. Probability distributions can be separated into two categories: discrete and continuous distributions [23, 24]. A discrete probability distribution is one that only takes a countable set of values. Continuous distributions, on the other hand, have a support (points where they are not zero) of uncountable size, e.g., the real numbers of zero to infinity. In quite a few cases, continuous distributions might be more suited to model a discrete problem while transforming the result back into the discrete space.

In this paper, we make heavy use of the normal distribution with the expected value $\mu$ and variance $\sigma^2$. The probability distribution is defined by its probability density function (PDF). The support, i.e., non zero values of the PDF, of the normal distribution is (−inf, inf)).

The support being all of the real numbers $\mathbb{R}$ can be problematic for some applications. To address this, the truncated normal distribution limits the distribution to some interval $[a, b] \subset \mathbb{R}$.

This requires re-normalisation of the result. As the integral of the PDF of the normal distribution over [a, b] is smaller than 1.

Another distribution based on the normal distribution is the log-normal distribution. Here, the domain is transformed, changing the default support to (0, +inf). This distribution is useful when the logarithm of the data is normally distributed.

Lastly, we make use of the Weibull distribution as a representative of the extreme value distributions family. This family models maxima or minima, with a support of [0, +inf).

## 2.6 Adaptive diffusion

Adaptive diffusion [19] is a protocol developed to address the privacy impact of the symmetry of flood-and-prune broadcasts. It breaks the symmetry present in regular flood-and-prune broadcasts by creating a virtual, or fake, source of the message. The virtual source spreads the message in such a way that they are the Jordan centre of the graph of nodes that received the message so far, not the true source of the message.

The virtual source cannot stay with the true source. To move the virtual source away, the current virtual source designates a neighbour as the new virtual source probabilistically. The goal is to equalise the probability of any node, that already received the message, to be the true source. In other words, if $n$ nodes received the message, the probability of any node $v_i$ having received the message being the true source $v$ should be approximately $P[v_i = v] \approx \frac{1}{n}$.

The forwarding probabilities are dependent on the underlying network assumptions. Adaptive diffusion uses a d-regular infinite tree as its basic network model, i.e., each node has exactly d neighbours, and there are infinite participants within the network. Further, message transmission happens only at discrete time steps. The message and message transmission are often called infections and may be used to model the spread of diseases. The probability of forwarding depends on the number of previous forwards $h$, the current number of steps so far $t$ and the degree of the underlying tree-network $d$. Using these parameters, the probability of designating a new virtual source is derived by Fanti et al. as

$$p_d(t, h) = \begin{cases} \dfrac{t - 2h + 2}{t + 2} & \text{if } d = 2, \\[2ex] \dfrac{(d-1)^{\frac{t}{2}-h+1} - 1}{(d-1)^{\frac{t}{2}+1} - 1)} & \text{if } d > 2. \end{cases} \tag{2}$$

Although this approach is designed for cycle-free networks, Fanti et al. provide [19] it works well even for general networks. For a network protocol, adaptive diffusion provides a few challenges. A suitably powerful attacker can subvert the protocol by connecting to a large number of nodes, as a node informs all neighbours of new messages, reducing the privacy guarantees to distance one. Further, later messages deliver the hop count $h$ of current forwards, eliminating all nodes of a distance not equal to $h$. Lastly, the discrete-time model needs to be transformed into a continuous-time model of real-world computing systems.

## 2.7 Alternative approaches

Bellet et al. [17] investigate a simple gossip protocol with a mute parameter $s$ over a complete graph, i.e., every node is connected to any other node. A node disseminating a message has a probability of $1 - s$ of stopping to disseminate after each interaction with a neighbour. With $s = 0$ a node forwards a message exactly once, while $s = 1$ leads to nodes forwarding the message via all available connections—essentially a flood-and-prune broadcast. Bellet et al. show that gossip with a mute value of $s$ has a differential privacy guarantee of $\delta = s + (1 - s)\beta$, where

$\beta$ is the fraction of attackers in the network, resulting in the lower bound of differential privacy equal to $\beta$. Our work in this paper is targeting different topologies, complicating the analysis.

Adaptive diffusion is not the only approach to privacy in contact networks or peer-to-peer networks. Dandelion [14] and Dandelion++ [15] attempt to provide privacy to Bitcoins peer-to-peer network. The protocol is based on an anonymity phase, where each node forwards the message exactly once, similar to the gossip protocol with the mute parameter equal $s = 0$. Unlike the gossip protocol, the selection of neighbours is not random, but follows a pattern: An approximation of a Hamilton path [14]. Nodes during this first phase have a chance, e.g., 10%, of switching to a flood-and-prune broadcast. One disadvantage of this behaviour is the indeterministic switch, prolonging the privacy phase for a long time, e.g., in the 95th percentile, a phase change requires more than 28 hops.

The main advantage of adaptive diffusion over the presented approaches is the non-probabilistic runtime guarantee and additional structure. This allows for deployment in systems with lower latency requirements [25].

## 3 k-growing $\eta$-adaptive diffusion

The general idea is still that of adaptive diffusion: The virtual source should forward messages so that it is the Jordan centre of the sub-graph created from all nodes that received the message. In detail, we apply some modifications to the protocol.

First, we limit the spread, i.e., the number of neighbours involved in the dissemination, to $\eta$ many neighbours. This change reflects in the message handling sub-protocol Algorithm 1, as nodes need to select a limited set of neighbours for a given protocol run, compare Line 2. We store the selected neighbours $\mathfrak{N}_m$ across multiple runs of the protocol, but for different messages $m$, the neighbours are selected again.

Further, arbitrary networks may have multiple paths between nodes, so a node may be selected as a neighbour for this protocol run, by multiple nodes. To prevent asymmetric spread, a node must only react on messages received via a single path. To enforce this, we store the first node we interact with given a message $m$ as the predecessor$_m$, see Line 3.

**Algorithm 1** $\eta$-adaptive diffusion message handling algorithm.

```
Input: Message m
Environment: Message sender v, Self v_self
1: if predecessor_m = ∅ then
2:    𝔑_m = randomly select η neighbours out of N(v_self)\{v}
3:    predecessor_m = v
4: else
5:    if predecessor_m = v then
6:       send m to all 𝔑_m
7:    end if
8: end if
```

The virtual source sub-protocol Aogorithm 2 requires further changes. The true source uses the message $(v, t = 1, r = \mathcal{H}(m))$ to initiate the protocol, which we call the virtual source token. $\mathcal{H}(m)$ is a suitable hash function to identify the current message efficiently. The hop counter $h$ has been dropped, as it leaks the distance to the true source to possible attackers.

On receiving the virtual source token, e.g., via Line 12, the recipient balances the spread of the message, so they are the centre of the spread graph. This is achieved by triggering the message handling algorithm on all neighbours, not including the node that sent the initiation message. This process is covered by lines 1 through 6.

The later part of the algorithm either forwards the message to all selected neighbours, see Line 16, which were selected in the message handling algorithm, Algorithm 1. Alternatively, the virtual source token is forwarded to a new virtual source ith probability $p_t$, c.f., Line 10.

The probability $p_t$ can be computed based on the distance distribution within the network $f$, i.e., $f(i)$ gives the expected number of nodes in distance $i$ of any node. The exact computation is quite involved; compare Section 4 for details.

**Algorithm 2** $\eta$-Adaptive Diffusion virtual source handling algorithm.

```
Input: Previous virtual source v_p, message identifier H(m), current
timestep t
Environment: Neighbours 𝔑_m with |𝔑_m| = η + 1, depth d
 1: for v ∈ 𝔑_m {v_p} do
 2:    Send m to v
 3:    if t + 1 ≤ d and t > 1 then
 4:      Send m to v
 5:    end if
 6: end for
 7: while t ≤ d do
 8:    t = t + 1
 9:    x =∼ 𝒰(0, 1)
10:    if x ≤ p_t then
11:       v_next =∼ 𝒰{𝔑_m {v_p}}
12:       Send (v_self, t, H(m)) to v_next, to call Algorithm 2
13:       break
14:    else
15:       Wait for ≈1 expected network latency
16:       for v ∈ 𝔑_m do
17:          Send m to v
18:       end for
19:    end if
20: end while
```

After a suitable threshold is reached for the privacy of the originator, i.e., the set of potential originators is large enough, the protocol switched to a flood-and-prune broadcast. This will ensure delivery to all participants and increase efficiency. At this point, privacy would barely improve by continuing adaptive diffusion. Expected privacy has reached its maximum once the full network is part of the set of potential originators.

To preserve the privacy of participants, nodes must monitor network latencies, as they have to artificially slow down the protocol when keeping a virtual source token. Further, every virtual source node must monitor the network for the progress of the protocol. A time-out will trigger retransmission to a different participant, as the previously selected is considered as refusing cooperation or unreachable. The time-out will extend when a message related to the current protocol instance is received, i.e., it concerns the message $m$. The time-out will only stop when receiving a flood-and-prune message relating to the same message $m$.

# 4 Privacy for general networks

## 4.1 Challenges

Some generalisations arise when considering general computer networks instead of infinite tree graphs. General networks may have cycles, i.e., multiple paths between participants, and a non-regular distance distribution. To extend the model of adaptive diffusion to these circumstances, we replace the calculations based on properties of a tree with a more general distribution function $f$, i.e., there are $f(i)$ nodes with distance $i$.

To prevent an attacker from learning additional information about the originator, we have to modify some aspects of the protocol. First, we have to remove the $h$ used in the protocol, as an attacker can infer the exact distance to the originator. As other participants may not know the distance distribution of the originator and to keep the protocol general, we will use a

homogeneous distribution, i.e., all nodes use the same distribution $f$ to compute their probabilities.

First, we will analyse the ideal situation for virtual source passing. Based on the results in an ideal setting, we show the minimal required modifications for non-ideal settings.

## 4.2 Ideal virtual source passing probabilities

As there is no fixed topology to analyse, we need to model the process of passing the virtual source token in a more abstract way. To model the process, we use a time inhomogeneous Markov chain, i.e., the probabilities involved may change based on the time $t$. For a network of diameter $\varnothing$ the chain has $\varnothing + 1$ states $0, 1, \ldots, \varnothing$. Each state represents the current distance of the virtual source from the true source.

A node of distance $h$ to the true source should pass the virtual source token to a more distant node with probability $p_t(h)$. Alternatively, the node keeps the distance the same with probability $1 - p_t(h)$. The Markov chain with these properties is visualised in Fig 2.

As noted before, a participant may not know its actual distance to the true source $h$, so the probabilities $p_t(h)$ may not depend on the distance to the true source.

At time $t$, let the $i$-th row of the vector $P_t \in [0, 1]^t$ describe the probability of the virtual source token being with a node of distance $i$ from the true source. We have $P_1 = (1)$, as the true source has distance 0 to itself and has the token initially. Further, let $M_t \in [0, 1]^{t+1 \times t}$ be the stochastic column matrix describing the transition from the $t$-th to the $(t + 1)$-st step, i.e., $P_{t+1} = M_t P_t = M_t M_{t-1} \ldots M_1 P_1$. Based on our Markov model, the matrix $M_t$ has the form:

$$M_t = \begin{pmatrix} 1 - p_t(0) & 0 & 0 & \cdots & & 0 \\ p_t(0) & 1 - p_t(1) & 0 & & & \vdots \\ 0 & p_t(1) & \ddots & \ddots & & 0 \\ \vdots & & & \ddots & \ddots & 0 \\ \vdots & & & & \ddots & 1 - p_t(t-1) \\ 0 & \cdots & & \cdots & 0 & p_t(t-1) \end{pmatrix}.$$

To solve for probabilities $p_t(h)$, we define our goal state: the probabilities for all any reachable node should be

$$\frac{1}{\#\text{reachable nodes in step } t} = \frac{1}{\sum_{s=0}^{t-1} f(t)}.$$

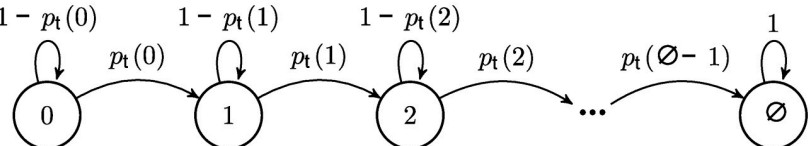

**Fig 2. Time inhomogeneous Markov chain of passing the virtual source token.**

Using this, we can describe the probability of a node of distance $i$ having the token at step $t$ as

$$\mathfrak{f}_t(i) = \frac{f(i)}{\sum_{s=0}^{t-1} f(s)}.$$

Using this, we can write the goal of equal probability as

$$P_t = \begin{pmatrix} \mathfrak{f}_t(0) \\ \mathfrak{f}_t(1) \\ \vdots \\ \mathfrak{f}_t(t-1) \end{pmatrix} = \frac{1}{\sum_{s=0}^{t-1} f(s)} \begin{pmatrix} f(0) \\ f(1) \\ \vdots \\ f(t-1) \end{pmatrix} \overset{!}{=} M_t M_{t-1} \ldots M_1 P_1.$$

Unfortunately, the number of restrictions does not necessarily allow for a single solution, perfectly fulfilling our goal. We can compute a possible solution $p_t$ based on the last row of our transition equation.

$$M_{t-1} P_{t-1} = \begin{pmatrix} 1 - p_{t-1}(0) & & & \\ p_{t-1}(0) & \ddots & & \\ & \ddots & 1 - p_{t-1}(t-2) & \\ & & p_{t-1}(t-2) & \end{pmatrix} \begin{pmatrix} \mathfrak{f}_{t-1}(0) \\ \mathfrak{f}_{t-1}(1) \\ \vdots \\ \mathfrak{f}_{t-1}(t-2) \end{pmatrix} = \begin{pmatrix} \mathfrak{f}_t(0) \\ \mathfrak{f}_t(1) \\ \vdots \\ \mathfrak{f}_t(t-1) \end{pmatrix} = P_t$$

First line:

$$(1 - p_{t-1}(0))\mathfrak{f}_{t-1}(0) = \mathfrak{f}_t(0)$$

$$\Leftrightarrow \quad 1 - p_{t-1}(0) = \frac{\mathfrak{f}_t(0)}{\mathfrak{f}_{t-1}(0)}$$

$$\Leftrightarrow \quad p_{t-1}(0) = 1 - \frac{\mathfrak{f}_t(0)}{\mathfrak{f}_{t-1}(0)}$$

$$\Leftrightarrow \quad p_t(0) = 1 - \frac{\mathfrak{f}_{t+1}(0)}{\mathfrak{f}_t(0)}$$

$(i + 1)$-st line:

$$\mathfrak{f}_t(i) = p_{t-1}(i-1)\mathfrak{f}_{t-1}(i-1) + (1 - p_{t-1}(i))\mathfrak{f}_{t-1}(i)$$

$$\Leftrightarrow \quad (1 - p_{t-1}(i))\mathfrak{f}_{t-1}(i) = \mathfrak{f}_t(i) - p_{t-1}(i-1)\mathfrak{f}_{t-1}(i-1)$$

$$\Leftrightarrow \quad p_{t-1}(i) = 1 - \frac{\mathfrak{f}_t(i) - p_{t-1}(i-1)\mathfrak{f}_{t-1}(i-1)}{\mathfrak{f}_{t-1}(i)}$$

$$\Leftrightarrow \quad p_t(i) = 1 - \frac{\mathfrak{f}_{t+1}(i) - p_t(i-1)\mathfrak{f}_t(i-1)}{\mathfrak{f}_t(i)}$$

By induction we arrive at:

$$\underbrace{\mathfrak{f}_t(i)p_t(i)}_{a_i} = \mathfrak{f}_t(i) - \mathfrak{f}_{t+1}(i) + \underbrace{p_t(i-1)\mathfrak{f}_t(i-1)}_{a_{i-1}}$$

$$\Leftrightarrow \qquad \mathfrak{f}_t(i)p_t(i) = \sum_{j=0}^{i}(\mathfrak{f}_t(j) - \mathfrak{f}_{t+1}(j))$$

$$\Leftrightarrow \qquad p_t(i) = \frac{\sum_{j=0}^{i}(\mathfrak{f}_t(j) - \mathfrak{f}_{t+1}(j))}{\mathfrak{f}_t(i)}$$

As the actual distance of a node from the origin is unknown, we have to determine a single probability. As the distribution over $h$ is known—it is our desired state $\mathfrak{f}_t$—we can combine these with the precomputed probabilities per distance. This achieves a single forwarding probability:

$$p_t = \sum_{h=0}^{t-1}\mathfrak{f}_t(h)p_t(h).$$

A node that did not forward the token could recompute the forwarding probability using its expected distance from the previous round to achieve better hiding.

## 4.3 Non-ideal virtual source passing

The ideal solution only holds if and only if the next state $P_t$ is reachable from $P_{t-1}$ by a single increase or stay. The condition can be formalised with the following requirements, derived from the solution:

$$0 \le \frac{\mathfrak{f}_t(0)}{\mathfrak{f}_{t-1}(0)} \le 1 \tag{3}$$

$$0 \le \frac{\sum_{j=0}^{i}(\mathfrak{f}_t(j) - \mathfrak{f}_{t+1}(j))}{\mathfrak{f}_t(i)} \le 1 \tag{4}$$

Eq (3) is always true by construction, as $f(i) > 0$ and

$$\frac{\mathfrak{f}_t(0)}{\mathfrak{f}_{t-1}(0)} = \frac{\frac{f(0)}{\sum_{s=0}^{t-1}f(s)}}{\frac{f(0)}{\sum_{s=0}^{t-2}f(s)}} = \frac{\sum_{s=0}^{t-2}f(s)}{\sum_{s=0}^{t-1}f(s)} \le 1.$$

Eq (4) intuitively describes that the probability of a node in distance $j$ possessing the token cannot exceed the probability of a node of the same distance possessing the token in the previous time step in addition to the total change in lower distances.

If this condition is violated, we need to compensate in the distribution or probabilities. Either way, the resulting distribution will be non-optimal hiding. To minimise the deviation, we determine the final desired state of the protocol, after $t$ steps, with $t \le \varnothing$. We then compute

a new $P_i'$, $\forall i \leq t$ as

$$P_i' = \begin{pmatrix} \mathfrak{f}_i'(0) \\ \vdots \\ \mathfrak{f}_i'(i-1) \end{pmatrix}$$

Here, $\mathfrak{f}'$ is derived from $\mathfrak{f}$ as:

$$\mathfrak{f}_t'(i) = \begin{cases} \mathfrak{f}_t(i) & \text{if } t \text{ is max desired state} \\ \mathfrak{f}_t(i) + \max\left(\chi_{t,i}, \delta_{t,i}\right) & \text{otherwise} \end{cases}$$

$$\chi_{t,i} = \sum_{j=i+1}^{t} \mathfrak{f}_t(j) - \mathfrak{f}_t'(j)$$

$$\delta_{t,i} = \mathfrak{f}_{t+1}'(t) - \mathfrak{f}_t(i) + \sum_{ji+1}^{t-1}(\mathfrak{f}_{t+1}'(j) - \mathfrak{f}_t'(j))$$

The value $\delta_{t,i}$ represents the difference required to fulfil Eq (4). On the other hand, $\chi_{t,i}$ represents all changes made to later entries, i.e., propagating the changes made through $\delta$. Note that Eq (4) is equivalent to the following.

$$0 \leq \frac{\sum_{j=0}^{i}(\mathfrak{f}_t(j) - \mathfrak{f}_{t+1}(j))}{\mathfrak{f}_t(i)} \leq 1$$

$$0 \leq \sum_{j=0}^{i}(\mathfrak{f}_t(j) - \mathfrak{f}_{t+1}(j)) \leq \mathfrak{f}_t(i)$$

Applying this equation to our goal state $\mathfrak{f}'$ we find the generation of $\mathfrak{f}'$ through the following changes:

$$\mathfrak{f}_t'(i) \geq \sum_{j=0}^{i}(\mathfrak{f}_t'(j) - \mathfrak{f}_{t+1}'(j))$$

$$= \sum_{j=0}^{i}\mathfrak{f}_t'(j) - \sum_{j=0}^{i}\mathfrak{f}_{t+1}'(j)$$

$$\stackrel{\sum_{j=0}^{k-1}\mathfrak{f}_k(j)=1}{=} 1 - \sum_{j=i+1}^{t-1}\mathfrak{f}_t'(j) - \left(1 - \sum_{j=i+1}^{t}\mathfrak{f}_{t+1}'(j)\right)$$

$$= \sum_{j=i+1}^{t-1}\mathfrak{f}_t'(j) + \sum_{j=i+1}^{t}\mathfrak{f}_{t+1}'(j)$$

$$= \sum_{j=i+1}^{t-1}\mathfrak{f}_t'(j) + \sum_{j=i+1}^{t-1}\mathfrak{f}_{t+1}'(j) + \mathfrak{f}_{t+1}'(t)$$

$$= \sum_{j=i+1}^{t-1}(\mathfrak{f}_t'(j) + \mathfrak{f}_{t+1}'(j)) + \mathfrak{f}_{t+1}'(t).$$

This leaves us with only known values, allowing us to compute the minimum difference required, i.e., $\delta_{t,i}$. We showed that if it is possible to achieve an optimal result, probabilities

derived from $f'$ are optimal. If such a result is not possible, probabilities derived from $f'$ will yield a result with minimum deviation for intermediate steps.

## 4.4 Continuous time

All previous discussions are in discrete time, i.e., the time $t$ is in steps, especially natural numbers. A network protocol must operate in some form of continuous-time or at discrete time-steps small enough to be considered continuous for practical purposes. Fortunately, network protocols lend themselves for a simple conversion technique: network latency.

If there was no delay between messages, a token transfer to another node could be observed by all participants of the protocol so far. To prevent this observation, a node must insert an artificial latency when not forwarding the message. The latency should be drawn from a distribution indistinguishable from real network latencies. Therefore, a node must observe the latencies of its connections.

## 4.5 Spread reduction

One remaining privacy problem of adaptive diffusion is the selection of all neighbours for dissemination. If an attacker is a neighbour of the first recipient of the virtual source token, they will notice the broadcast as soon as possible without being the first virtual source recipient. An attacker can force this situation by creating connections to all participants in the network. Even with many unobtrusive attackers distributed throughout the network, the chance of selection is high.

To reduce privacy leaks, we introduce the parameter $\eta$. Participants only select $\eta$ neighbours to participate in the protocol instead of all neighbours. This reduces the chance of selecting at least one attacker.

Limiting the set of participating neighbours prevents full coverage of the network through adaptive diffusion alone. Therefore, an additional flood-and-prune phase is necessary to ensure delivery to all network participants. Lastly, lower values of $\eta$ increase the required time to reach larger parts of the network, e.g., 21 nodes are reached after three spread rounds with $\eta = 2$, while $\eta = 5$ reaches 30 nodes in two spread rounds.

## 4.6 Limitations

Unfortunately, a node cannot reliably decide which edges increase or decrease the distance to the true source, as the source is unknown or an edge with the desired probability is not available. For every node, keeping the token will keep the distance the same. As a heuristic for early nodes, returning the token along the path it was received likely reduces the distance by one, while forwarding it to another node likely increases the distance. Knowledge about the neighbours of neighbours can increase the accuracy of this heuristic.

For networks observed in real-world peer-to-peer-networks, the small-world property likely holds [22, 26]: the shortest distance between any two nodes is likely below or equal to 6. After six steps, the candidate pool for the true source is most of the network. Therefore, performing the analysis is sufficient for the early steps of the protocol.

Due to the lack of information stemming from the privacy requirements, the distribution of nodes holding the virtual source is distorted at every step, making the result less accurate. Alternative approximations based on the distribution may perform better empirically. One improvement may be a better approximation by nodes holding the virtual source token. They can infer their distance to the true source to be at most $t$ the moment they receive the token. Therefore, they have no reason to use the probabilities as if they were at distance $t + 1$ should they keep the token.

Lastly, the previous section's result is based on a distribution of the shortest paths within the networks. This distribution is not generally known for most graph types and could not be empirically determined by a participant without knowledge of the topology.

## 5 Distribution model

We analysed expected k-growing network topologies, which are similar to real-world peer-to-peer network growth, for their distance distributions. This relieves the final limitation, knowledge about a concrete distribution. The result allows a node to compute $p_t$ based on the number of expected edges per node and the number of nodes in the network.

### 5.1 Distributions for alternative models

Fronczak et al. [27] derive an exact solution for random Erdös-Rényi graphs, i.e., random graphs where all edges are equally likely. They especially consider Erdös-Rényi graphs with two hidden variables $h_i$ and $h_j$. Let $\gamma \approx 0.5772$ be Euler's constant, the resulting average degree distribution is given by

$$l = \frac{-2(\ln h) + \ln N + \ln(h^2) - \gamma}{\ln N + \ln(h^2) - \ln \beta} + \frac{1}{2}. \tag{5}$$

Loguinov et al. [28] investigate structured peer-to-peer networks. They provide a succinct overview over shortest path results for Chord and CAN networks. Chord shortest paths are binomially distributed, which tends to a normal distribution for larger values, while CAN becomes normal as well, for increasing CAN dimensions. Lastly, they propose an architecture using de Bruijn graphs, which have no closed-form for their distribution of shortest paths but give an exponential approximation.

Roos et al. [29] derive a model for Kademlia like systems. Kademlia is a structured peer-to-peer network with routing based on $b$-bit long identifier spaces. They consider a network of order $n$, where routing considers $\alpha$ close nodes to reach $\beta$ nodes close to the target, based on a $k$-bucket routing system. They model the hop count distribution of a given system via a Markov chain. They derive a space complexity of $\mathcal{O}\left(\frac{b^{2\alpha}}{(\alpha!)^2}\right)$ and computation complexity of $\mathcal{O}(nb^{\alpha(\beta+2)})$ for their resulting model. As $\alpha, \beta$ and $k$ are usually constant for a given deployment, it is manageable for network participants but not optimal. For details about the fairly complex model, refer to the original publication [29].

The results provided by these works are applicable if the network conforms to the proposed structure. Unfortunately, they do not map well to the proposed model, lack an approach for parameter inference and are complex to use.

### 5.2 Methodology

We determine suitable distributions and their parameters by creating and analysing random graphs. To create the graphs, we use igraph, while the analysis is done using scipy [30].

We chose igraph's graph establishment function, which takes a number of nodes $n$ and a number of edges per node $k$. The method creates a random graph by sequentially adding nodes. Each node creates $k$ edges to already existing nodes. This scheme leads to a connected graph, where older nodes have a higher number of connections, while new nodes have at least $k$ connections.

We chose this scheme as it is similar to the schemes used in peer-to-peer networks. A new node connects to publicly known nodes and asks for a set of participants. The new node then chooses some number of nodes to connect to. This model is a simplification, as it ignores

churn, i.e., nodes leaving and joining the network again, but it is a close fit for real-world applications.

To reproduce the steps and results of this paper, we provide a repository of our data and scripts under the MIT open source license, including an interactive notebook for experimentation: https://github.com/vs-uulm/eta-adaptive.

## 5.3 Models for distance distribution

To model the observed behaviour, we chose various discrete and continuous distributions. As discrete candidates, we looked at Poisson, Planck, Binomial and Geometric distributions. For continuous distributions, we considered the normal, log-normal, truncated normal and Weibull distribution. We evaluated the fit of continuous distributions by the overall shape, as the data is discrete.

The Weibull distribution was chosen as a candidate for the extreme value distributions, as shortest paths are calculated as minimums over paths. The normal distribution was chosen due to the central limit theorem, i.e., the normal distribution as the limit of independent samplings. The log-normal and truncated normal distributions were selected as a candidate as its support can be limited to positive values—a sensible limitation for path lengths.

We estimated parameter fits for all distributions from many generated graphs. The parametrisation for the truncated normal was mostly indistinguishable from the produced normal distribution. Similarly, the log-normal distribution was transformed to mimic the normal fit closely. Therefore, we removed the truncated and log-normal distribution as candidates to not overcomplicate the model. Representative examples for continuous fits are shown in Fig 3.

Similarly, the results for the discrete distributions was not a good fit. Only the Poisson distribution produced a convincing fit for any graph but limited to graphs with $k = 1$. We did not

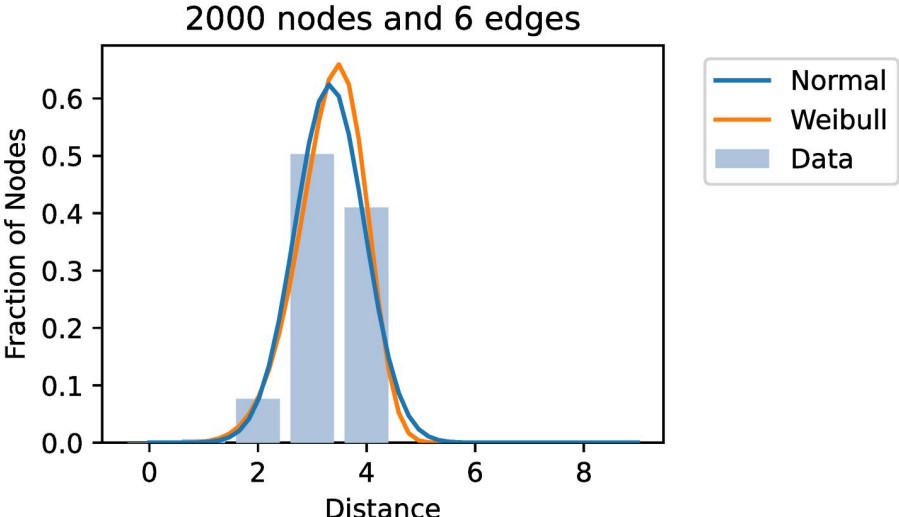

**Fig 3. Fitted continuous distributions from an example dataset, which was created using 2000 nodes and 6 edges per node.** A lognormal and truncated normal fit were plotted identically to the normal distribution and were, therefore, omitted.

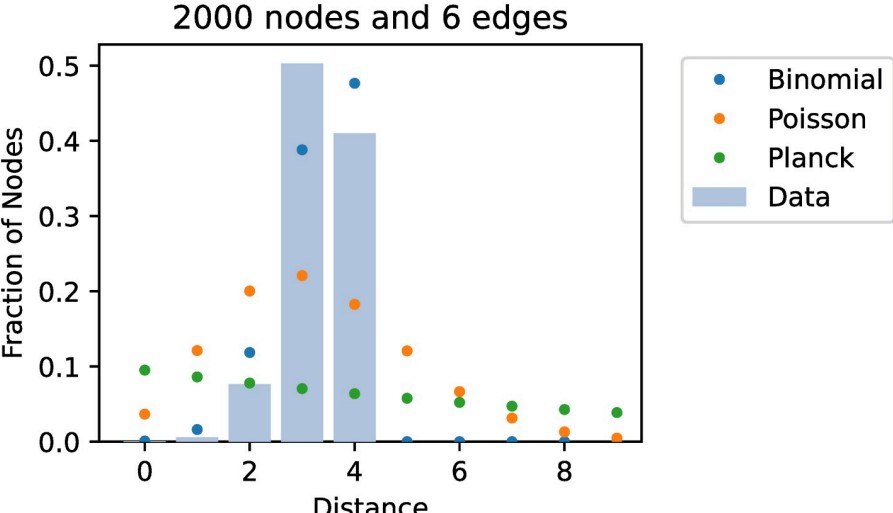

**Fig 4. Fitted discrete distributions from an example dataset, which was created using 2000 nodes and 6 edges per node.** Only the binomial estimation using the ceiling operator to discretize the parameters shows any resemblace to the desired data.

test other discrete distributions as often no efficient maximum likelihood estimators exist or are implemented. Representative examples for discrete fits are shown in Fig 4.

Finally, the results for the normal distribution produced good point-wise fits for graphs with $k > 1$. The normal distribution fits were especially accurate for the core section of the distribution, which is also consistent with findings of normally distributed path lengths in other peer-to-peer networks [28]. The most significant deviation from the data could be observed for the low end of the distribution: for distance 0 or 1. Fortunately, these values can be fitted manually based on the parameters $n$, $k$, as the mass at a distance of zero should be $\frac{1}{n}$ and the expected mass at the distance of one should be $\frac{k(2n-k-1)}{n^2}$, i.e., the average degree.

### 5.4 Discretization

To apply the normal distribution to our given problem, the resulting distribution needs to be discretised, i.e., turned from a continuous distribution into a discrete one. The main goal is to keep the properties of a probability distribution, i.e., the sum of all points not equal to zero needs to add up to 1.

A valid discretization can be constructed based on the cumulative distribution function (CDF) over intervals, capturing the full support of the distribution, e.g., $f(x) = \text{CDF}(x)$

As we fit the distribution based on the points of data, the natural discretisation can be achieved by point-wise evaluation and re-normalisation of the result. Let PDF be the probability density function, then a new probability mass function with evaluation points 0, 1, . . ., $t$ (the discrete equivalent to a PDF) is given by $x \in \{0, 1, . . ., t\}$

$$f_t(x) = \frac{\text{PDF}(x)}{\sum_{s=0}^{t} \text{PDF}(s)}.$$

This approach can easily accompany special values at certain points. Therefore, the full discretization of our normal distribution shall be:

$$
\begin{aligned}
f(0) &= \frac{1}{n} \\
f(1) &= \frac{k(2n - k - 1)}{n^2} \\
f(x) &= \frac{\mathrm{PDF}(x)}{f(0) + f(1) + \sum_{s=2}^{t} \mathrm{PDF}(s)}.
\end{aligned}
$$

The maximum point $t$ should be chosen in such a way, that the remaining error $1 - \mathrm{CDF}(t) \leq \epsilon$ is small enough for the given purpose.

## 5.5 Model parameter estimator

The previous section concluded that the distribution of shortest paths could be modelled using a discretised normal distribution. Building upon this conclusion, we are further interested in the parameters $\mu$ and $\sigma^2$ of a normal distribution $\mathcal{N}(\mu, \sigma^2)$. The parameters of the normal distribution should only depend on the parameters of our network, the number of nodes $n$ and number of edges $k$. We are interested in functions $M, S$, with a small error err such that

$$
\mu = M(n, k) + \mathrm{err}
$$
$$
\sigma = S(n, k) + \mathrm{err}.
$$

These are statistical estimators. To determine these, we fitted a large number of randomly generated graphs and stored the resulting values for $\mu$ and $\sigma$. The determined functions $M, S$ are approximated using the functional equations:

$$
M(n, k) \approx \frac{\alpha \ln(\beta n)}{e^{\gamma k}} + \delta \ln(\eta n) + \frac{\zeta}{e^{\gamma k}} + \epsilon,
$$
$$
S(n, k) \approx \mathfrak{a} \ln(\mathfrak{b} n) + \frac{\mathfrak{c}}{e^{\mathfrak{d} k}} + \mathfrak{e}.
$$

Here, the greek and fracture constants $\alpha$ to $\epsilon$ and $\mathfrak{a}$ through $\mathfrak{e}$ are determined by least-square fitting of the function to the acquired data. Through a fit of experimental data, we reached the following approximate functions:

$$
M(n, k) \approx \frac{0.595 \log(2.135n)}{\exp(0.314k)} + 0.341 \log(1.626n) + \frac{0.241}{\exp(0.314k)} - 0.224,
$$
$$
S(n, k) \approx 0.0345 \log(0.925n) + \frac{1.222}{\exp(0.301k)} + 0.189.
$$

## 5.6 Landscape

We used the fitted parameters for random graphs to determine the behaviour of the parameters. The ranges of the parameters depend on the size of the network $n$ and the number of connections created in each step $k$.

By splitting the dimensions based on $n$ and $k$, an initial estimation is possible. The dimension dependent on $k$ shows a behaviour proportional to $\frac{1}{e^k}$. The dimension dependent on $n$, on the other hand, shows a behaviour proportional to $\log(n)$, a square root behaviour could be

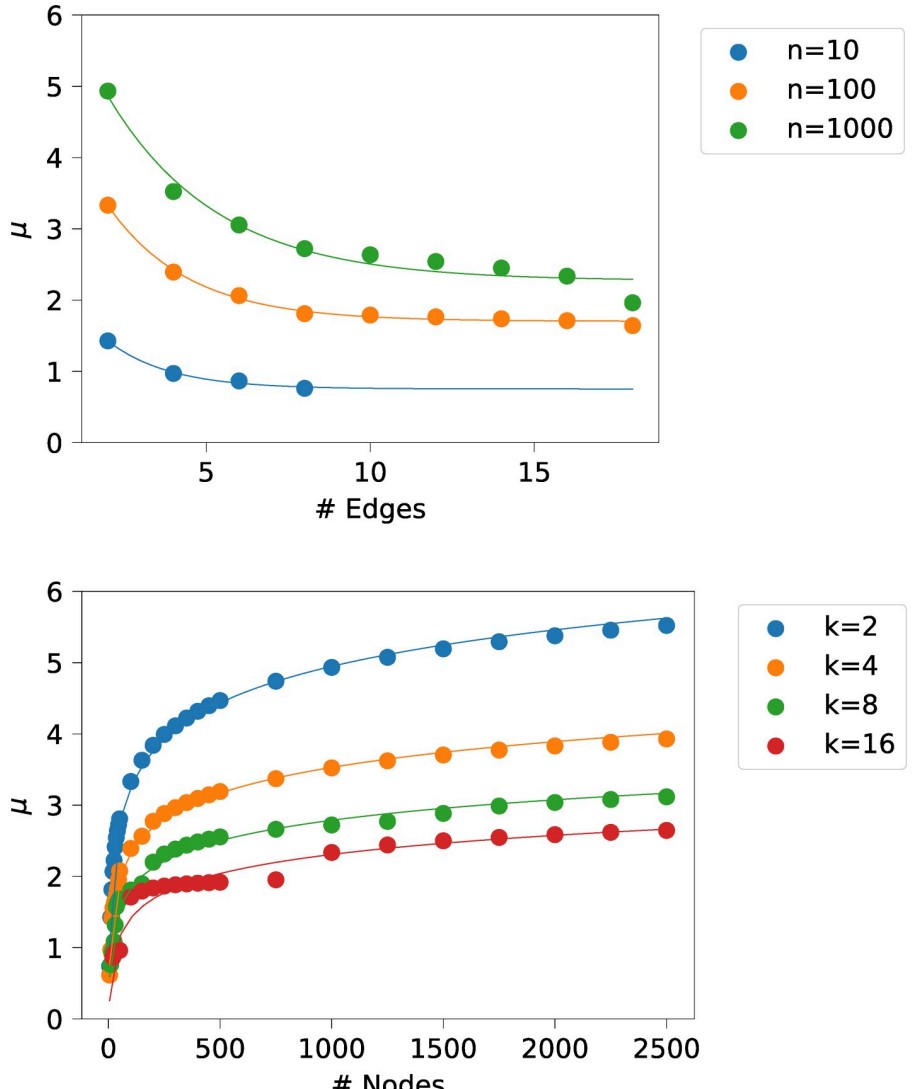

**Fig 5. Datapoints for various graph sizes split by number of edges per node $k$ and number of nodes $n$.** $\mu$ estimate fitted using $\frac{1}{e^k}$ for the number of edges per node and fitted using $\log(n)$ for the number of nodes dimension.

excluded as fitted parameters easily overestimated the data. A random selection of the dimensional analysis is shown in Fig 5.

The estimation of values for $\sigma$ show much more pronounced residues in the form of a sawtooth function. The forms can be recognised from the similarly shaped but smaller residues of the $\mu$ estimations. The values of $\sigma$ show to be within 0.3 to 0.7, even for large numbers of nodes $n$, e.g., $n = 1\,000\,000$. Further, the values seem to jump rapidly and slowly descend, forming a sawtooth pattern, which is hard to predict accurately. The pattern arises as additional nodes in the network are more likely to create shortcuts than to increase path length until the overall network diameter increases by one, steeply widening the distribution—and therefore increasing the variance, i.e., $\sigma$. The likelihood of such an increase follows its own probability distribution, which we did not determine for this paper.

### 5.7 Estimator models

Based on our one dimensional evaluation, we want to construct a two dimensional estimator model $M(n, k)$. Model candidates are based on possible combinations of our one dimensional approximations, i.e., the partial derivatives are the derivatives of our observations:

$$\frac{\partial M}{\partial n} \approx \frac{d}{dn} \log(n),$$

$$\frac{\partial M}{\partial k} \approx \frac{d}{dk} \frac{1}{e^k}.$$

The constructed models have various constants, denoted by greek lower case symbols. These constants are not shared between the models but were independently fitted. The models are denoted by

$$M_1(n, k) = \alpha \ln(\beta n) + \frac{\gamma}{e^{\delta k}} + \epsilon,$$

$$M_2(n, k) = \frac{\alpha \ln(\beta n)}{e^{\gamma k}} + \delta \ln(\eta n) + \frac{\zeta}{e^{\gamma k}} + \epsilon,$$

$$M_3(n, k) = \frac{\alpha \ln(\beta n)}{e^{\gamma k}} + \epsilon,$$

$$M_4(n, k) = \frac{\alpha \ln(\beta n)}{e^{\gamma k}} - \frac{\alpha \delta k}{e^{\gamma k}} + \frac{\zeta}{e^{\eta k}} + \frac{\theta \ln(\iota n)}{(\kappa n)^{\lambda n}} + v \ln(\xi n).$$

We fit the parameters of the model based on our first dataset. The residuals of the fit parameters, i.e., the difference between the true value and estimation, shows a sawtooth form. This arises as additional nodes likely create shortcuts until the overall diameter of the network grows.

To evaluate the performance besides this observed error, we created a new independent dataset. We measured the difference between the calculated values by our model and the actual fitted parameters. This difference corresponds to the bias of the estimator, which is simply referred to as bias. Fig 6 shows the distribution of the bias of our models for $\mu$.

For the $\mu$ estimation, model $M_2$ and $M_4$ perform the best. Model $M_2$ requires less parameters than $M_4$, i.e., it is simpler, therefore we prefer model $M_2$.

For the estimation of $\sigma$, none of the models performs exceptionally well. In general, the observed sigma values are small and within the range 0.3-0.7, even for graphs using one million nodes. As no model performed exceptionally well, but also not exceptionally bad, we stuck with the simplest model. The simplest model based on number of parameters is model 1. Fig 7 provides a discretization of an example prediction for a dataset based on 2000 nodes and 6 edges per node.

We can now use these values to compute concrete $p_t$ for a given network of $n$ nodes using a $k$-growing approach.

## 6 Privacy discussion

In this section, we investigate the privacy impact of the protocol. The main challenge for quantification is the arbitrary topology and topology abstractions.

### 6.1 Model

Given a network of size $n$, we have a set of attackers $A$ of size $|A|$ participating in the network; therefore, $n - |A|$ is the number of all fully honest nodes. The value $\beta = \frac{|A|}{n}$ represents the

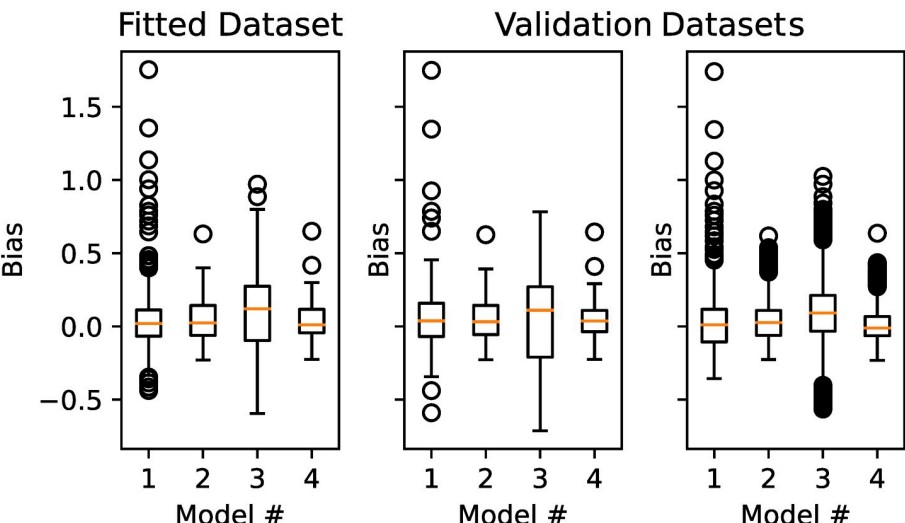

**Fig 6. Boxplot of the bias distribution of the $\mu$-estimator, using the four models, compared to the measured value.**

fraction of attackers within the network, which is the probability of selecting an attacker when selecting a node from the network uniformly at random. Lastly, any node has an expected number of connections $c > 1$ to other nodes.

To locate a node in a flood-and-prune broadcast, an attacker node registers an incoming message through one of its $c$ connections. Given the knowledge of the topology, the attacker can separate the network in $c$ many sets of nodes, where any node is in the same set, if a broadcast from this node would reach the attacker via this connection. These sets are not necessarily disjunct, as there may be multiple shortest paths between a node and the attacker. To find a lower bound on the privacy effects, we assume the sets are disjunct, as this improves the position of the attackers.

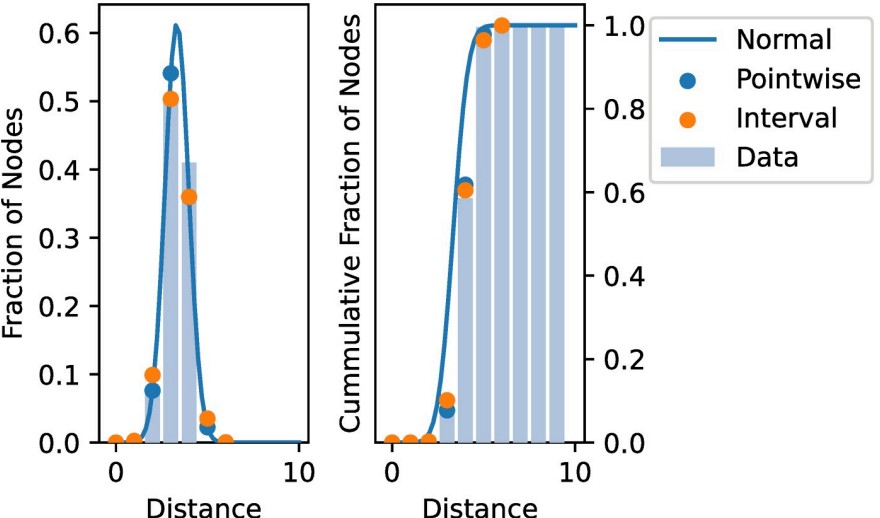

**Fig 7. Discretization of an example dataset using 2000 nodes and 6 edges per node.** The predicted normal distribution is based on parameters estimated using model 2 for $\mu$ and model 1 for $\sigma$, with a pointwise discretization and an interval discretization, using midpoint intervals.

The expected size of such a set for a single attacker node will be $\frac{n-|A|}{c}$, as an attacker would not consider colluding attackers. For this model, we consider only untargeted attacks, where attackers try to deanonymise any sending node, not a single particular node. In this case, the ideal set of network nodes any attacker can distinguish based on a message is $\frac{n-|A|}{c}$. Further, ideal sets for attackers minimise the size of multiple observing attackers to the size of $\frac{n-|A|}{c^{|A|}}$.

Attackers can fully deanonymise a sending node once this size is below one. It follows that attackers can correctly determine the location of a sender in this ideal setting, once more than $\log_c(n-|A|)$ attackers have received the message, as it holds, that

$$\frac{n-|A|}{c^{|A|}} \leq 1 \Leftrightarrow |A| \geq \log_c(n-|A|). \tag{6}$$

## 6.2 Spreading sub-protocol

Given the required number of attackers, we are interested in when this number is likely reached during the spreading protocol. Nodes that select neighbours may select nodes that already participate in the protocol, so the tree sub-graph resulting from participating nodes is not complete.

If all nodes select their neighbours uniformly at random from the full set of network participants, this becomes a form of the coupon collectors problem in packs [31]. The coupon collectors problem using packs describes the problem of receiving at least one of each coupon when receiving coupons in packs of a given size. Here, the size of the packs is given by $\eta$, and each network node represents a coupon.

Given a subset $A$ of all coupons, the random variable $Z_\ell(A)$ is the number of drawings necessary to obtain at least $\ell$ elements of $A$ [31]. The expected value of reaching the required number of attackers $Z_{\log_c(n-|A|)}(A)$ can be calculated using the results of Stadje:

$$E(Z_{\log_c(n-|A|)}(A)) =$$
$$\binom{n}{c}^{\lceil \log_c(n-|A|) \rceil - 1} \sum_{j=0}^{} \frac{(-1)^{\lceil \log_c(n-|A|) \rceil - j + 1}}{\binom{n}{c} - \binom{n-|A|+j}{c}} \binom{|A|}{j} \binom{|A|-j-1}{|A|-\lceil \log_c(n-|A|) \rceil}. \tag{7}$$

The number of drawings corresponds to the number of nodes participating in the protocol before reaching the required number of attackers. To calculate a lower bound of the depth of the $\eta$-adaptive diffusion dissemination tree at this point, we invert the calculation of the number of nodes in a complete tree.

$$Z_{\log_c(n-|A|)}(A) \geq 1 + (\eta+1)\sum_{i=0}^{t-2} \eta^i \tag{8}$$

$$\Leftrightarrow t \geq \log_\eta\left(1 - \frac{(Z_{\log_c(n-|A|)}(A))(1-\eta)}{\eta+1}\right) + 1 \tag{9}$$

Table 2 provides an overview of the evaluation of this function when 5% of the network is colluding. It shows that for low values of $\eta$ a significant depth can be reached, as four steps in a network with an average degree of $c = 8$ results in a candidate pool for the true source for approximately $c^4 = 8^4 = 4096$. The likelihood of any of the candidate nodes being the true source depends on the distribution $\mathfrak{f}$ of Section 4.

**Table 2. Expected tree depth for an attacker fraction of $\beta = 0.05$ before deanonymization of the virtual source.**

| n / $\eta$ | 100 | 1000 | 10000 |
|---|---|---|---|
| 3 | 4.3 | 3.9 | 7.8 |
| 5 | 2.5 | 2.7 | 2.7 |
| 10 | 1.6 | 1.7 | 1.9 |

## 6.3 Virtual source sub-protocol

A node received additional information when chosen as the virtual source. An attacker is chosen as the virtual source with probability

$$P[\text{virtualsource} \in A] = \frac{|A|}{n-1}. \tag{10}$$

In this formulation, a successful selection occurs when the virtual source is an attacker. This is a simple Bernoulli trial, so the geometric distribution gives the expected number of trials until a success occurs as

$$E(P[\text{virtualsource} \in A]) = \frac{n-1}{|A|} \approx \frac{1}{\beta}. \tag{11}$$

For a reasonable network size ($n > 100$) this value stays above 6 for fractions of attackers below $\beta = 0.166$. This result is expected, as the process of virtual source selection mimics the privacy-optimal process of the work by Bellet et al. [17] with a muting parameter of $s = 0$ and earlier similar results.

## 7 Future work

There are various smaller improvements possible to increase the privacy or efficiency of the protocol. First, instead of switching to a flood-and-prune from the last virtual source node, the protocol could instead trigger the flood-and-prune broadcast from all leaf nodes. The last message transmission message, see Algorithm 1, would instruct leaf nodes to start a flood-and-prune process. This would reduce leaks of information during the flood-and-prune protocol and improve the efficiency of the protocol.

To improve resistance against linkable broadcasts and to hinder an attacker in the first step, the current timestep $t$ may be randomised on initiating. This would also require the originator to use a spreading message first, as the initial transmission would otherwise be special, as a node should only receive the virtual source after receiving a spreading message.

The protocol has little guards against non-participation attacks and communication failures, which could be mitigated through retransmissions and time-outs. While allowing the protocol to complete, these would still reduce the efficiency of the protocol in selective non-participation attacks. As those do not prevent every connected node from receiving the message and do not diminish the privacy results, we did not tackle these in this paper.

Lastly, a more extensive privacy analysis would benefit the protocol. Due to the lack of topology information, the privacy analysis is limited in its applicability. A more accurate result could be achieved for specific topologies or considering distributions over topologies.

## 8 Conclusion

In this paper, we transformed the adaptive diffusion protocol [19] into a realistic protocol for current peer-to-peer networks. To achieve this, we remodelled the virtual source passing probabilities in a more general way, based on the distance distribution of the underlying network and improved the attacker model by removing information from protocol messages. Further, we provide a privacy-friendly solution to solve these equations while smoothing out otherwise unachievable states.

We analysed expected k-growing network topologies, which are similar to real-world peer-to-peer network growth, for their distance distributions. The analysis showed the distances in the networks to be approximately normally distributed. Lastly, we performed a parameter analysis of the resulting normal distributions, showing that $\mu$ and $\sigma$ of the normal distribution can be approximated by a combination of logarithmic and inverse exponentials. The models, based on the number of edges per node $k$ and number of nodes $n$ are:

$$\mu(n, k) \approx \frac{0.595 \log (2.135n)}{\exp(0.314k)} + 0.341 \log (1.626n) + \frac{0.241}{\exp(0.314k)} - 0.224$$

$$\sigma(n, k) \approx 0.0345 \log (0.925n) + \frac{1.222}{\exp(0.301k)} + 0.189$$

All results are available online to evaluate and reproduce under a permissive open source license: https://github.com/vs-uulm/eta-adaptive.

## Author Contributions

**Conceptualization:** David Mödinger.

**Data curation:** David Mödinger.

**Formal analysis:** David Mödinger, Jan-Hendrik Lorenz.

**Investigation:** David Mödinger, Jan-Hendrik Lorenz, Franz J. Hauck.

**Methodology:** David Mödinger, Jan-Hendrik Lorenz, Franz J. Hauck.

**Resources:** David Mödinger.

**Software:** David Mödinger.

**Supervision:** Franz J. Hauck.

**Writing – original draft:** David Mödinger.

**Writing – review & editing:** David Mödinger, Jan-Hendrik Lorenz, Franz J. Hauck.

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
