## [Decision Letter · Decision Letter 0]

17 Feb 2021

PONE-D-20-35748

Statistical privacy preserving message dissemination for peer-to-peer networks

PLOS ONE

Dear Dr. Mödinger,

Thank you for submitting your manuscript to PLOS ONE. After careful consideration, we feel that it has merit but does not fully meet PLOS ONE’s publication criteria as it currently stands. Therefore, we invite you to submit a revised version of the manuscript that addresses the points raised during the review process.

We look forward to receiving your revised manuscript.

Kind regards,

Chakchai So-In, Ph.D.

Academic Editor

PLOS ONE

Journal Requirements:

Reviewers' comments:

Reviewer's Responses to Questions

**Comments to the Author**

1. Is the manuscript technically sound, and do the data support the conclusions?

Reviewer #1: Yes

Reviewer #2: Yes

2. Has the statistical analysis been performed appropriately and rigorously? 

Reviewer #1: Yes

Reviewer #2: Yes

3. Have the authors made all data underlying the findings in their manuscript fully available?

Reviewer #1: Yes

Reviewer #2: Yes

4. Is the manuscript presented in an intelligible fashion and written in standard English?

Reviewer #1: Yes

Reviewer #2: Yes

5. Review Comments to the Author

Reviewer #1: This paper transforms the adaptive diffusion protocol in the reference into a protocol for

peer-to-peer networks. To achieve this, the authors remodel the virtual source passing

probabilities in a more general way, based on the distance distribution of the underlying

network. Further, the paper provides a privacy-friendly solution to solve these equations, while

smoothing out otherwise unachievable states.

In general, the problem investigated in the manuscript is interesting. The paper might be improved by addressing the following issues.

1. Some of the sentences presented is confusing. For example, in the abstract, what does "optimal privacy" and "limited information environment" mean?

2. In introduction, all the references are related to cryptocurrencies. Can you provide other practical practical examples on the possible usage of the proposed protocols considering the title of the paper seems to be applicable to more general networks systems and applications.

3. The introduction part can be more clear. For example, regarding the last sentence on section 2, is it possible to discuss a little bit more on why it is not suitable? Due to what kind of constraints or settings?

4. In section 3.1, can we show a simple motivated example that the originator can be found easily?

5. For section 3.3, it is suggested to make it concise and even remove this.

6. It is suggested to put notations in a list or table for reference to be more clear. Also, some terms are not defined before using them. For example, "v_{p,m}" in Algorithm 3, and "r = H(m)" were not defined.

7. In the algorithms, it might be more clear to put output at the end of the algorithm. So that it is more clear where the algorithm starts.

8. Considering the practical communication networks, what is the impact of transmission failures to the proposed protocol? Also, what about packet loss? Or other constraints in practical networks.

9. Is it possible to quantify the level of privacy with the dependence of other factors in the proposed protocol?

Reviewer #2: This paper discusses the privacy of broadcasts within an unstructured p2p network and proposes a protocol to ensure the anonymity of the source of the broadcast.

My main concern are:

- peer-to-peer networks are also characterized by dynamics with churn and evolution of the topology. For instance, protocols such as Gossip-based Peer Sampling manage to ensure only one connected component, a random topology, and a strong resilience to churn. In this article, the considered topology does not take into account churn and the analysis is performed on a static topology which seems unrealistic for a peer-to-peer network. If the considered dissemination is not affected by churn for some reasons (very fast / instantaneous dissemination), please clarify it.

- I have a problem with this statement (which justifies the use of a k-growing model as proposed): "Peer-to-peer networks are modeled as organically growing graphs". Please support your claim with references. Depending on the underlying overlay construction to manage the network, this claim is not right. If you consider specific networks where this claim is right, please specify.

Other comments:

- The introduction is very short, please extend it to better introduce the context / existing limitations / the main idea of the paper and why it is relevant. The term one-dimensional random-walk is mentioned in the contributions but does not appear in the rest of the paper, please improve the consistency of used terms. Section 6 is not introduced in the roadmap.

- The paper does not analyse the impact of colluding nodes. Simulating a growing rate of colluding nodes and an associated sensitivity analysis could be interesting.

- The architecture of Section 6 is strange, there is no sense to have only one subsection.

- Section 3.4: Adaptive diffusion creates a virtual source token. To better understand, can you quickly explain how this virtual source is created.

- Algorithms should be better explained by using lines as reference.

- Section 4: "a node must only react on messages received via the proper path." Please, clarify what you refer to “proper path”.

- Recent works partly address similar problems, please consider them.

@misc{bellet2020started,

title={Who started this rumor? Quantifying the natural differential privacy guarantees of gossip protocols},

author={Aurélien Bellet and Rachid Guerraoui and Hadrien Hendrikx},

year={2020},

eprint={1902.07138},

archivePrefix={arXiv},

primaryClass={cs.DC}

}

@inproceedings{decouchant:hal-02421820,

TITLE = {{P3LS: Plausible Deniability for Practical Privacy-Preserving Live Streaming}},

AUTHOR = {Decouchant, J{\\'e}r{\\'e}mie and Boutet, Antoine and Yu, Jiangshan and Esteves-Verissimo, Paulo},

URL = {https://hal.inria.fr/hal-02421820},

BOOKTITLE = {{SRDS 2019 - 38th International Symposium on Reliable Distributed Systems}},

ADDRESS = {Lyon, France},

PAGES = {1-10},

YEAR = {2019},

MONTH = Oct,

}

6. PLOS authors have the option to publish the peer review history of their article (what does this mean?). If published, this will include your full peer review and any attached files.

Reviewer #1: No

Reviewer #2: No

---

## [Author Response · Author response to Decision Letter 0]

17 Mar 2021

Dear Reviewer #1,

thank you for your extensive review of our manuscript.

We would like to let you know that we addressed the issues your review raised in the following way:

1. Some of the sentences presented is confusing.

 We tried to reword larger parts of the manuscript to make the meaning more clear.

2. In introduction, all the references are related to cryptocurrencies.

 We added flooding content lookup in p2p filesharing applications as examples to the introduction.

 Files and content can reveal a lot about individuals simply from the metadata of files or keywords requested.

3. The introduction part can be more clear.

 We extended and rewrote parts of the introduction, e.g., clarifying why we think the original model is not suitable for real world computer networks.

4. In section 3.1, can we show a simple motivated example that the originator can be found easily?

 We added an additional example, including a short graphic, to illustrate the problem.

5. For section 3.3, it is suggested to make it concise and even remove this.

 We shortened section 3.3 (due to restructures no section 2.5) by removing explicit equations and only keeping more terse descriptions.

6. It is suggested to put notations in a list or table for reference to be more clear. Also, some terms are not defined before using them. For example, "v_{p,m}" in Algorithm 3, and "r = H(m)" were not defined.

 We added a short notation overview and tried to make notation less ambigious, e.g., by renaming v_p,m in the algorithm to "predecessor_m".

7. In the algorithms, it might be more clear to put output at the end of the algorithm. So that it is more clear where the algorithm starts.

 Unfortunately, we did not follow this suggestion.

 Although we understand the motivation, there is no easy to define output, as the algorithm are more akin to reactions on received messages.

8. Considering the practical communication networks, what is the impact of transmission failures to the proposed protocol? Also, what about packet loss? Or other constraints in practical networks.

 We added a paragraph on the shortcomings in these regards, as the selective non-participation measures also apply to transmission failures.

 Although, it is advisable to either use a transport protocol that mitigates these issues or implement additional measures to prevent unnecessary performance drops.

9. Is it possible to quantify the level of privacy with the dependence of other factors in the proposed protocol?

 We added a new section (6) to discuss the privacy protperties.

 The section discusses the expected length of uncaptured virtual source messages and the extent to which the protocol may run before attackers capture enough information to identify the virtual source reliably.

 We give some example results for a network with 5% attackers.

Please let us know if you recommend further changes to improve the mansucript!

Dear Reviewer #2,

thank you for your extensive review of our manuscript.

We would like to let you know that we addressed the issues your review raised in the following way.

Regarding your main concerns:

 * peer-to-peer networks are also characterized by dynamics with churn and evolution of the topology. For instance, protocols such as Gossip-based Peer Sampling manage to ensure only one connected component, a random topology, and a strong resilience to churn. In this article, the considered topology does not take into account churn and the analysis is performed on a static topology which seems unrealistic for a peer-to-peer network. If the considered dissemination is not affected by churn for some reasons (very fast / instantaneous dissemination), please clarify it.

 We rewrote and extended the background discussion of networks models to clarify our model and why we decided to not take churn into account in the network model.

 While the dissemination is fast, network issues may raise problems.

 These can be countered by measures against (selective-)non-participation, e.g., a leaving node may be treated as malicious.

 (The challenge of freshly joining nodes depends on the decision if they should receive the message or may be ignored.)

 * I have a problem with this statement (which justifies the use of a k-growing model as proposed): "Peer-to-peer networks are modeled as organically growing graphs". Please support your claim with references. Depending on the underlying overlay construction to manage the network, this claim is not right. If you consider specific networks where this claim is right, please specify.

 We clarified that our network model and analysis is restricted to a specific form of unstructured peer-to-peer network and added alternative models for distance distributions.

 Further, we added references for various known distributions of structured and unstructured networks as alternatives to our analysis in suitable contexts.

 Lastly, we added references to a description of the Bitcoin network, which originally inspired this construction.

Regarding your other comments:

 * The introduction is very short, please extend it to better introduce the context / existing limitations / the main idea of the paper and why it is relevant. The term one-dimensional random-walk is mentioned in the contributions but does not appear in the rest of the paper, please improve the consistency of used terms. Section 6 is not introduced in the roadmap.

 We extended the introduction a bit and tried to improve the wording and description of the problem.

 For example, we removed terms that are not used in the rest of the manuscript and instead use more consistent wording.

 We updated the roadmap to reflect all sections 2 to 6 properly.

 * The paper does not analyse the impact of colluding nodes. Simulating a growing rate of colluding nodes and an associated sensitivity analysis could be interesting.

 As we strongly agree with this, we introduced a new section 6 to discuss some privacy aspects of the protocol, but further analysis is of interest.

 Unfortunately, some obstacles make the desired analysis very extensive, so we also added additional investiations to the future work section.

 * The architecture of Section 6 is strange, there is no sense to have only one subsection.

 We merged the old section 6 into the previous section, as it fits better with the overall discussion of (now) section 4.

 * Section 3.4: Adaptive diffusion creates a virtual source token. To better understand, can you quickly explain how this virtual source is created.

 We removed the notion of "virtual source token" from the description of adaptive diffusion, as the notion was not correct in this way.

 We added the description of what we call the virtual source token at the apropriate place in the main section 3.

 ("the message (v; t = 1; r = H(m)) [...], which we call the virtual source token").

 * Algorithms should be better explained by using lines as reference.

 We added explanations for central aspects of the resulting algorithms by line reference, as suggested.

 To improve the section on adaptive diffusion, we removed the original algorithm and focues the section on the concept.

 * Section 4: "a node must only react on messages received via the proper path." Please, clarify what you refer to “proper path”.

 We rephrased the sentence (and other parts of the neighbouring paragraphs) to make the meaning of a single return path more clear.

 * Recent works partly address similar problems, please consider them.

 We considered both refered works and added Bellets work as well as other works we were familiar with, but did not discuss in the manuscript previously.

 We limited the discussion to similar protocols though, even though there are further privacy protocols with vastly different dynamics (e.g., protocols based on dining cryptographer networks) to preserve some focus.

 After careful consideration we die not include P3LS as we believe it tackles a very different problem, as it is grounded in video stream distribution instead of message distribution.

Please let us know if you recommend further changes to improve the mansucript!

Thank you and regards,

David Mödinger

---

## [Decision Letter · Decision Letter 1]

27 Apr 2021

Statistical privacy-preserving message broadcast for peer-to-peer networks

PONE-D-20-35748R1

Dear Dr. Mödinger,

We’re pleased to inform you that your manuscript has been judged scientifically suitable for publication and will be formally accepted for publication once it meets all outstanding technical requirements.

Kind regards,

Chakchai So-In, Ph.D.

Academic Editor

PLOS ONE

Additional Editor Comments (optional):

Reviewers' comments:

Reviewer's Responses to Questions

**Comments to the Author**

1. If the authors have adequately addressed your comments raised in a previous round of review and you feel that this manuscript is now acceptable for publication, you may indicate that here to bypass the “Comments to the Author” section, enter your conflict of interest statement in the “Confidential to Editor” section, and submit your "Accept" recommendation.

Reviewer #1: All comments have been addressed

Reviewer #2: All comments have been addressed

2. Is the manuscript technically sound, and do the data support the conclusions?

Reviewer #1: Yes

Reviewer #2: Yes

3. Has the statistical analysis been performed appropriately and rigorously? 

Reviewer #1: Yes

Reviewer #2: Yes

4. Have the authors made all data underlying the findings in their manuscript fully available?

Reviewer #1: Yes

Reviewer #2: Yes

5. Is the manuscript presented in an intelligible fashion and written in standard English?

Reviewer #1: Yes

Reviewer #2: Yes

6. Review Comments to the Author

Reviewer #1: The presentation might be improved further. All other questions are addressed. The reviewer does not have any other questions for this manuscript.

Reviewer #2: Thank you for your revision, you fully answered my comments and clarified raised concerns.

This new version is much better.

7. PLOS authors have the option to publish the peer review history of their article (what does this mean?). If published, this will include your full peer review and any attached files.

Reviewer #1: No

Reviewer #2: No

---

## [Editor Report · Acceptance letter]

28 Apr 2021

PONE-D-20-35748R1 

Statistical privacy-preserving message broadcast for peer-to-peer networks 

Dear Dr. Mödinger:

I'm pleased to inform you that your manuscript has been deemed suitable for publication in PLOS ONE. Congratulations! Your manuscript is now with our production department. 

Kind regards, 

on behalf of

Dr. Chakchai So-In 

Academic Editor

PLOS ONE